# Improving Mask R-CNN for Nuclei Instance Segmentation in Hematoxylin & Eosin-Stained Histological Images

**Benjamin Bancher**                                    benjamin.bancher@gmail.com
and **Amirreza Mahbod**                         amirreza.mahbod@meduniwien.ac.at
and **Isabella Ellinger**                        isabella.ellinger@meduniwien.ac.at
*Institute for Pathophysiology and Allergy Research, Medical University Vienna, Vienna, Austria*

**Rupert Ecker**                                    rupert.ecker@tissuegnostics.com
*Department of Research and Development, TissueGnostics GmbH, Vienna, Austria*

**Georg Dorffner**                              georg.dorffner@meduniwien.ac.at
*Section for Artificial Intelligence, Medical University of Vienna, Austria*

**Editor:** TBA

## Abstract

Digital pathology is an emerging topic in the analysis of pathologic tissue samples. It includes providing the tools towards more automated workflows to derive clinically relevant information. Digitization and storage of whole slide images have become commonplace and allow modern image analysis methods to be used. In recent years, computer-based segmentation of cell nuclei has gathered considerable attention in the development of highly specialized algorithms. Currently, most of these algorithms are based on performing semantic segmentation of all cell nuclei and separating overlapping instances in a post-processing step. Recently, instance-aware segmentation methods such as Mask R-CNN have been proposed to enable unified instance detection and segmentation, even in overlapping cases. In this work, we propose a modified Mask R-CNN-based approach by incorporating distance maps of instances and hematoxylin-stain intensities as extra input channels to the model. Moreover, we explore the impact of three well-known inference strategies, namely test-time augmentation, ensembling, and knowledge transfer through pre-training on the segmentation performance. We perform extensive ablation experiments across multiple runs to quantitatively define the most optimal inference strategy in the proposed Mask R-CNN algorithm. Our results show that average instance segmentation improvements of up to 3.5% and 4.1% based on Aggregate Jaccard Index and Panoptic Quality score can be obtained, respectively, using the proposed techniques in comparison to a standard Mask R-CNN model. Our findings confirm the effectiveness of aggregating information at the network input stage and optimizing inference workflows using minimal effort. Implemented modifications and codes are publicly available through a GitHub repository under: https://github.com/bbanc/Improved-Mask-R-CNN-for-nuclei-segmentation.

**Keywords:** Mask R-CNN, Nuclei Segmentation, Medical Image Analysis, Deep Learning, Computational Pathology

## 1. Introduction

Digital pathology has a wide range of applications in the computer-based analysis of histological samples. For instance hematoxylin and eosin (H&E-) stained images are routinely digitized to be more easily accessible and to allow the use of image analysis methods to facilitate diagnosis (Pallua et al. (2020); Zarella et al. (2019)). These semi- or fully-automatic

methods can be used for various applications including nuclei instance segmentation. Nuclei density, count, and the ratio of nucleus to cytoplasm are some of the main features used when performing cancer grading and deriving treatment plans in routine pathology (Aeffner et al. (2019); Skinner and Johnson (2017)). Performing accurate and automatic nuclei instance segmentation in digitized histological images is a challenging task due to several reasons. The existence of touching, or even heavily overlapping nuclei, inconsistent nuclei size and shape, and significant variations in color distributions of nuclei in various samples are among the most common issues for the development of any automatic nuclei segmentation model (Kumar et al. (2017); Salvi et al. (2021)). Besides these challenges, to use supervised machine learning approaches, ground truth data at pixel level are required to train the models. Deriving these ground truth annotations is a non-trivial task due to both inter- and intra-observer variability in manually created annotations (Aeffner et al. (2017); Kumar et al. (2017, 2020); Mahbod et al. (2021)).

Recently, deep learning-based image analysis methods have shown promising success in the field. Moreover, multiple nuclei instance segmentation (or classification) challenges such as MoNuSeg2018 (Kumar et al. (2020)), Kaggle DSB2018 (Caicedo et al. (2019)), and MoNuSAC2020 (Verma et al. (2021)) challenges helped to create publicly available datasets and standard benchmarking for nuclei instance segmentation. In these challenges, highly specialized methods have been proposed to bridge the gap between purely semantic and instance-based segmentation methods. Many of these approaches are based on a deep learning encoder-decoder-based semantic segmentation algorithm and aim to generate additional information to then separate overlapping nuclei in the final inference. Different approaches to models include regression-based models (Graham et al. (2019); Naylor et al. (2019); Mahbod et al. (2019)) or ternary segmentation-based models (Chen et al. (2016); Oda et al. (2018); Zhou et al. (2019)).

In contrast, instance aware-based models directly detect and segment nuclei. Mask R-CNN (He et al. (2020)), one of the most well-known detection-based models, is comprised of two main stages. The first performs object detection and localization, and the second stage uses the features of the detected regions of interest to jointly perform classification, final localization, and segmentation. A few adapted Mask R-CNN-based approaches have been introduced in the literature for nuclei instance segmentation. Johnson (2018) adapted Mask R-CNN for the Kaggle DSB2018 and noted the impact of optimized training procedures and the use of deeper network backbones. MACD-RCNN (Ma et al. (2020)) is an adaptation for abnormal nuclei detection using fixed-size region proposals and an attention mechanism. Nuclei R-CNN (Lv et al. (2019)) employs Mask R-CNN using a tile-based instance merging approach to infer on large image patches. Vuola et al. (2019) proposed ensembling Mask R-CNN with U-Net as a way of harnessing the strengths of each of the two architectures. Recently, Mask R-CNN has also been shown to significantly benefit from the use of test time augmentation for instance segmentation (Moshkov et al. (2020)).

We modified the original Mask R-CNN architecture (He et al. (2020)) for the nuclei instance segmentation task. Our main contributions can be summarised as follows:

- We propose a modified Mask R-CNN-based approach by incorporating the distance maps of instances and hematoxylin-stain (H-stain) intensities as extra input channels.

- We exploit three well-known inference strategies, namely test-time augmentation (TTA), ensembling, and knowledge transfer through pre-training to improve the segmentation performance.

- We showcase an optimized use of Mask R-CNN in nuclei segmentation with minimal addition to the original Mask R-CNN architecture.

- We provide quantitative comparisons of the baseline Mask R-CNN architecture to its improved versions.

- We perform extensive ablations to accurately measure the impact of various changes on the Mask R-CNN segmentation performance.

## 2. Methods

### 2.1. Datasets

To assess the quantitative performance of Mask R-CNN and its modified versions, we used the MoNuSeg2018 dataset (Kumar et al. (2017)). This dataset is composed of 30 training and 14 test images with close to 22,000 training and 7,000 test nuclei instances. All images are H&E-stained, acquired at 40× magnification, and are fully manually annotated. The dataset is composed of 1000 × 1000 pixel image tiles extracted from whole slide images of nine different biopsy sites (breast, kidney, liver, prostate, bladder, colon, and stomach in the training set; breast, kidney, prostate, bladder, colon, lung, and brain in the test set).

In addition to the MoNuSeg2018 dataset, model pre-training and knowledge transfer were explored through the PanNuke dataset (Gamper et al. (2019, 2020)). PanNuke is a semi-automatically annotated segmentation and classification dataset consisting of over 200,000 nuclei across 19 different organ sites (bladder, ovaries, pancreas, thyroid, liver, testes, prostate, stomach, kidney, adrenal gland, skin, head and neck, cervix, lung, uterus, esophagus, bile-duct, colon, and breast) at 40× magnification. It contains more than 7,000 image patches with a fixed size of 256 × 256 pixels. Further details of the MoNuSeg2018 and PanNuke datasets can be found in their respective publications.

### 2.2. Modifying Mask R-CNN

An emerging theme in current literature is the use of additionally derived information such as distance (Graham et al. (2019); Mahbod et al. (2019)) or hematoxylin channel (H-channel) (Zhao et al. (2020)) information to improve model performance. Inspired by these trends, we propose to extend Mask R-CNN by adding extra input channels to allow the model to make use of H-stain intensity information and estimated distance maps derived from a Euclidean distance transform. All our experiments were based on the open-source Matterport implementation of Mask R-CNN (Abdulla (2017)).

H-stain intensity was estimated by using the color deconvolution approach proposed by (Ruifrok et al. (2003)). Inspired by the Beer-Lambert law of light absorption it aims to estimate the relative contribution of a certain stain in an RGB image and separates the various stain contributions to the observed color using color deconvolutions. It captures the contributions of the hematoxylin component of the stain and therefore yields a clear intensity map encoding the positions and shapes of nuclei within the original RGB-image.

The Euclidean distance map was estimated using a distance U-Net similar to the model proposed in (Mahbod et al. (2019)). In analogy to (Mahbod et al. (2019)), we used a network with five max-pooling layers in the encoder part and five transposed convolutional layers in the decoder part. Moreover, we used dropout layers between all convolutional layers in the extracting and expanding paths with a probability of 0.1. Except for the last layer where a linear activation function was used, ReLu activations were utilized for all other convolutional layers. The distance map captures the distance of a pixel to its nearest nonzero pixel in a binary mask. Previous research has shown such contextual information to be a powerful tool in nuclei instance segmentation.

The additional estimated H-channel and Euclidean distance information were added as prior information to the RGB channels of the input image. This five-channel input data was then projected onto a standard three-channel input using a $1 \times 1$ convolutional layer. We refer to this extension of Mask R-CNN as Mask R-CNN+ henceforth. The generic flowchart of Mask R-CNN+ is depicted in Figure 1.

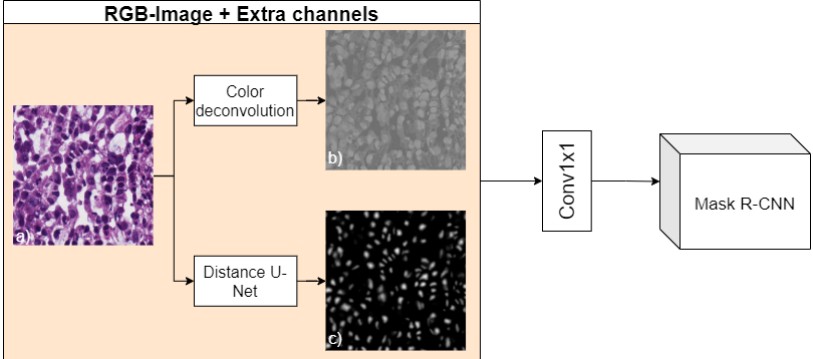

Figure 1: Generic workflow of Mask R-CNN+, a) RGB-image, b) extracted H-stain intensity, c) estimated distance map. During pre-processing, additional information is derived via extracted H-stain intensities and euclidean distance maps.

## 2.3. Incorporating three inference strategies

We explored and quantitatively assessed the impact of three inference strategies on the Mask R-CNN+ nuclei instance segmentation performance. We used TTA, ensembling, and knowledge transfer through pre-training.

Both TTA and ensembling are multiple inference methods and require their respective outputs to be merged into a single unified inference. This is a non-trivial task in instance segmentation (Kumar et al. (2020)). Traditionally, it is performed using a non-maximum suppression scheme. Non-maximum suppression in instance segmentation can be formulated as an algorithm, which first identifies which segmentations belong to the same detection (e.g. according to ground truth criteria, such as using an intersection over union metric), then uses a score to rank the matching segmentations and finally removes all but the best scoring one. We adapted this algorithm to suppress all singular detections (e.g. during ensembling a detection is suppressed if it is only seen by a single model) and merged all segmentations belonging to the same detection using average voting, as is commonly used in semantic segmentation ensembling.

Ensembling was set up to be organ-based by splitting the full MoNuSeg2018 training dataset into folds by their primary tumor site. This resulted in five distinct models trained on a combination of breast, kidney, liver, prostate, and a mixture of bladder, colon, and stomach samples. Their inferences were then aggregated using our adapted instance merging scheme. The overview of the utilized ensembling approach is shown in Figure 2.

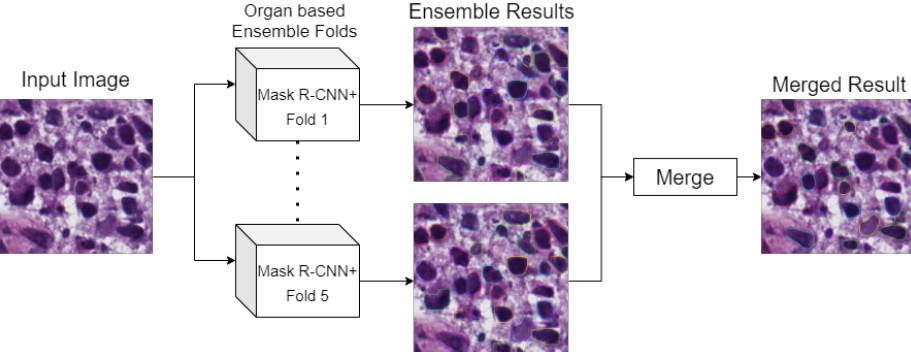

Figure 2: Overview of the utilised ensemble strategy. For ease of viewing only the RGB input channels are shown.

Recently, TTA has been shown to consistently improve model performance in nuclei segmentation (Moshkov et al. (2020)). Here, TTA was performed by augmenting the original input three times. Horizontal and vertical flips, as well as a Hue saturation augmented image were used as augmented images. The resulting instance segmentations were then merged using our adapted instance merging scheme. An overview of TTA is depicted in Figure 3.

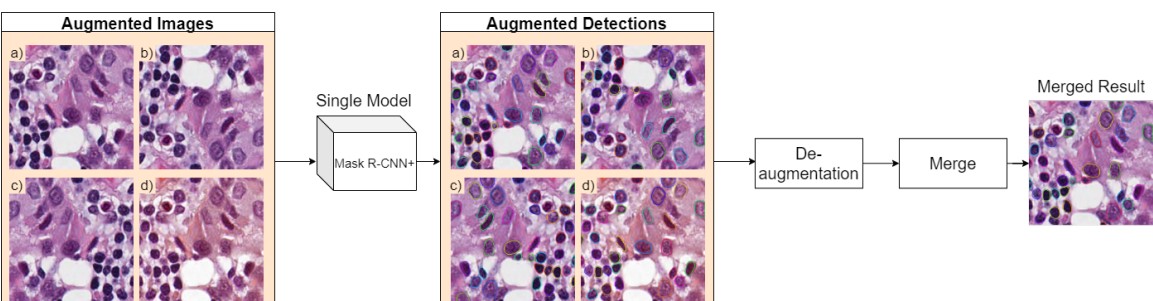

Figure 3: Test-time augmentation overview. For ease of viewing only the RGB input channels are shown. a) RGB-input, b) horizontal flipped image, c) vertical flipped image, d) hue augmented image

Model pre-training and knowledge transfer were explored through the PanNuke dataset. To make full use of the information provided, we trained a standard Mask R-CNN model on the joint segmentation and classification task. Model weights acquired through pre-training on the natural image COCO (Lin et al. (2014)) dataset were further trained for 30 epochs on the entire PanNuke dataset and later fine-tuned on the MoNuSeg2018 training set. The overall pre-training and fine-tuning scheme is depicted in Figure 4

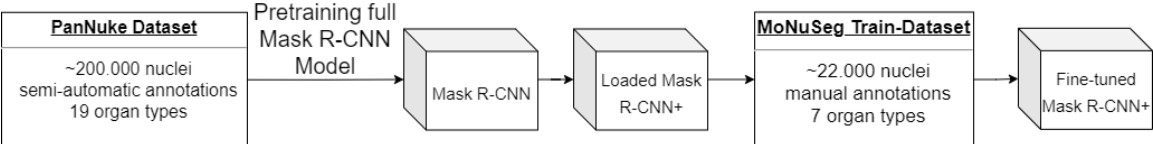

Figure 4: Pre-training and fine-tuning scheme.

## 2.4. Experimental Setup

To measure the instance segmentation performance, we used Aggregate Jaccard Index (AJI), Panoptic Quality (PQ) score, Detection Quality (DQ) score (or F1), Segmentation Quality (SQ) score (average SQ of all true positive detections), and standard Dice score (Kumar et al. (2017); Kirillov et al. (2019)). In order to capture model performance and measure variability in training we performed the same experiment five times and report the average results across these runs.

To train the distance U-Net model (see Section 2.2), we used randomly initiated weights using the He normal initializer (He et al. (2015)). As we dealt with a regression task, we used a mean square error loss function. Adam optimizer (Kingma and Ba (2015)) was employed to train the network. The initial learning rate was set to 0.001 with a learning scheduler that divided the learning rate by 10 after every 25 epochs. We trained the model for 80 epochs with a batch size of 2 to fit the full resolution images to the model. We used the entire MoNuSeg2018 training set for training and predicted the distance maps for the MoNuseg2018 test images.

To train Mask R-CNN-based models, we used specialized training schedules as suggested in (Hollandi et al. (2020)). The utilized training schedule for Mask R-CNN and Mask R-CNN+ targets different stages of the network at different stages of the training. We adopted this idea and therefore trained our models in four stages: The first stage only tunes the $1 \times 1$ input convolutional layer and output layers while keeping the backbone fixed to enable knowledge transfer. The second is the main training stage optimizing all network parameters. The final two stages consist of fine-tuning the final back bone-stages and output layers, and finally training the output layers only.

For all Mask R-CNN models, we used a network pre-trained on natural images from the COCO dataset (Lin et al. (2014)). The loss was the same as proposed in the original Mask R-CNN implementation (He et al. (2020)). We used an Adam optimiser (Kingma and Ba (2015)) with an initial learning rate of 0.0001. For optimized learning, this learning rate was multiplied by 2, 1, 0.5, and 0.1 for their respective learning stages. The ground truth distance maps were used during Mask R-CNN training. During inference, the estimated distance maps were combined with the standard input data and extracted H-channel information for the five-channel image input. Data augmentation during training included random brightness contrast changes, random rotations, as well as horizontal and vertical flips, and random crops of $256 \times 256$ pixels to fit the GPU-memory. We performed a Wilcoxon signed-rank test to define statistical significance between our best implementation and Mask R-CNN (Gibbons and Chakraborti (2011)).

## 3. Results

We propose an improved Mask R-CNN-based approach by incorporating distance and H-stain intensities and offer comparisons to a baseline Mask R-CNN approach. To evaluate the impact of our alternate inference strategies, we first perform singular tests and then used extensive ablations to define the best combination for the optimized segmentation performance. In all experiments, the models were trained on the MoNuSeg2018 training set and evaluated on the test set. We also provide the average values for seen and unseen organ types separately to better assess the model's generalization capabilities in the Appendix.

Table 1 shows the average results across five runs of the proposed methods and inference adaptations on the entire test set in terms of AJI, PQ, DQ, SQ, and DICE scores. Figure 6 shows a graphic representation of these results. The best method shown in bold was found to be statistically different to Mask R-CNN using the Wilcoxon signed-rank test (AJI: $p = 0.002$, PQ: $p = 0.001$, DQ: $p = 0.008$, SQ and DICE: $p < 0.001$).

| Methods/Score (%) | AJI | PQ | DQ/F1 | SQ | DICE |
|---|---|---|---|---|---|
| Standard Mask R-CNN | 60.97 | 60.49 | 82.20 | 73.47 | 77.87 |
| Mask R-CNN+ | 62.29 | 60.67 | 81.14 | 74.69 | 79.67 |
| Mask R-CNN+ \w Ensemble | 63.11 | 61.73 | 80.93 | 76.22 | 80.37 |
| Mask R-CNN+ \w TTA | 62.40 | 60.88 | 81.34 | 74.76 | 79.57 |
| Mask R-CNN+ \w PanNuke | 63.08 | 61.77 | 82.01 | 75.26 | 79.96 |
| Mask R-CNN+ \w PanNuke + Ensemble | 64.02 | 62.41 | 81.79 | 76.25 | 80.76 |
| Mask R-CNN+ \w PanNuke + TTA | 63.49 | 63.05 | 83.16 | 75.76 | 80.15 |
| Mask R-CNN+ \w Ensemble + TTA | 63.38 | 63.44 | 83.01 | 76.37 | 80.52 |
| Mask R-CNN+ \w Ensemble + TTA + PanNuke | **64.43** | **64.60** | **84.53** | **76.37** | **80.91** |

Table 1: Average experimental results on the MoNuSeg2018 test set across five runs, the best combination in terms of AJI for comparison with the MoNuSeg2018 results was found to include a combination of all proposed inference improvements (shown as bold).

## 4. Discussion

From the results in Table 1, we conclude that providing additional information and adapting inference methods in Mask R-CNN improves performance in terms of all measured scores. In the Appendix, Figure 6 shows considerable variability in model performance due to random initialization of untrained weights, random batches, and augmentation effects on training convergence.

In terms of adapted inference methods, we observed improvements across all explored modalities. A combination of model pre-training, knowledge transfer, and TTA were found to be the most impactful adaptation to our inference scheme. Ensembling through utilizing a set of organ-based folds and TTA both require multiple inference and merging of detected nuclei at the instance level. Out of these two methods, ensembling increased the instance segmentation performance more drastically, both in terms of AJI and PQ score. The PQ score is most notably improved by using multiple inference methods.

The best performing model of the MoNuSeg2018 challenge achieved an AJI of 69.07%. This is a superior result to our proposed optimized method, which reached an average score of 64.43%. To test for statistical significance we used a Wilcoxon signed-rank test and found our method to be statistically different on average. The superior methods in the MoNuSeg2018 challenge include highly specialized algorithms and deep learning architectures, such as the winner of the challenge (Zhou et al. (2019)). Their architecture introduces several new and highly adapted workflows to reach a final instance segmentation result, including customized loss functions to better incorporate the rare highly complex cases, new encoder block paths, and information aggregation modules combined in a novel architecture. Our scores, however, were achieved using Mask R-CNN with minimal adaptations to the original architecture and an optimized inference strategy. It is worth mentioning that the reported inter-observer variability of the challenge was 65.3% (95%CI 63.9-66.7), which is very close to the best results derived in our proposed approach.

To further improve the segmentation performance, a number of additional techniques can be used, that are not addressed in this study. These include adapting Mask R-CNN's loss functions to better reflect the unique complexities in nuclei segmentation. Specialized approaches for semantic segmentation networks include penalizing sharp edges with high curvature to help segment common round nuclei shapes (Wang et al. (2020)) or using an adapted loss function to put more emphasis on difficult or rare sample cases (Zhou et al. (2019)).

Graham et al. (2019) propose a tiling-based strategy to perform inference on overlapping image tiles. During this process, a large input image is tiled into multiple overlapping smaller ones and only the detection in the non-overlap regions are kept during inference. This, however, was found to not increase model performance in our experiments.

Another highly debated topic in the analysis of histological images is the use of stain normalization or color normalization. Salvi et al. (2021) concluded an improvement of model performance for the use in classification and detection tasks. Although in this work, we did not perform extensive experiments for color normalization, a standard stain normalization technique (Macenko et al. (2009)) did not yield a performance improvement.

Inference time is a metric not included in our experiments. It is however highly important when deploying segmentation algorithms into a digital pathology-based workflow. For this consideration, multiple inference techniques, such as ensembling or TTA need to be carefully considered, as they may lead to a significant increase in the inference time on the same hardware. However, if additional hardware is available, multiple inference lends itself well to parallelization leaving only the result aggregation as a final task.

## 5. Conclusion

Mask R-CNN is a powerful algorithm for nuclei instance segmentation. It can strongly benefit from additional information at the input stage and optimized inference workflows such as TTA, ensembling, and pre-training. Further research is needed to continue to explore and improve the performance of detection and instance-aware models such as Mask R-CNN.

## 6. Acknoledgements

The authors would like to thank the Austrian Research Promotion Agency (*Forschungsförderungsgesellschaft - FFG*) for funding this research as part of the *Deep nuclei analysis* project under FFG No. 872636. The authors are grateful for the help of the Research & Development team at TissueGnostics GmbH.

## Appendix A. Appendix

Table 2 shows the generalization capabilities of Mask R-CNN+ split on seen and unseen test organs.

| | AJI | PQ | DQ/F1 | SQ | DICE |
|---|---|---|---|---|---|
| Standard Mask R-CNN | | | | | |
| Seen average | 60.78 | 60.69 | 82.09 | 73.80 | 77.75 |
| Unseen average | 61.44 | 59.99 | 82.48 | 72.66 | 78.18 |
| Mask R-CNN+ | | | | | |
| Seen average | 61.39 | 60.02 | 80.08 | 74.82 | 79.24 |
| Unseen average | 64.55 | 62.30 | 83.79 | 74.37 | 80.75 |
| Ensemble | | | | | |
| Seen average | 61.99 | 60.98 | 80.03 | 76.10 | 79.78 |
| Unseen average | 65.91 | 63.61 | 83.17 | 76.53 | 81.87 |
| TTA | | | | | |
| Seen average | 61.30 | 60.21 | 80.73 | 74.52 | 79.03 |
| Unseen average | 65.12 | 64.27 | 83.83 | 76.58 | 81.59 |
| PanNuke | | | | | |
| Seen average | 62.33 | 61.31 | 81.22 | 75.39 | 79.60 |
| Unseen average | 64.94 | 62.92 | 83.97 | 74.94 | 80.85 |
| Ensemble + Pannuke | | | | | |
| Seen average | 63.08 | 61.84 | 81.03 | 76.23 | 80.22 |
| Unseen average | 66.37 | 63.85 | 83.71 | 76.30 | 82.10 |
| Ensemble + TTA | | | | | |
| Seen average | 62.28 | 62.72 | 82.13 | 76.28 | 79.94 |
| Unseen average | 66.12 | 65.23 | 85.19 | 76.62 | 81.97 |
| TTA + PanNuke | | | | | |
| Seen average | 62.66 | 62.50 | 82.38 | 75.78 | 79.72 |
| Unseen average | 65.58 | 64.41 | 85.11 | 75.70 | 81.22 |
| Ensemble + PanNuke + TTA | | | | | |
| Seen average | 63.52 | 64.04 | 83.80 | 76.34 | 80.39 |
| Unseen average | 66.69 | 66.02 | 86.35 | 76.47 | 82.19 |

Table 2: Generalization capabilities of Mask R-CNN+ on the MoNuSeg2018 test set based on seen (bladder, breast, kidney, colon and prostate, n = 10) image types and unseen ones (Brain, Lung, n = 4)

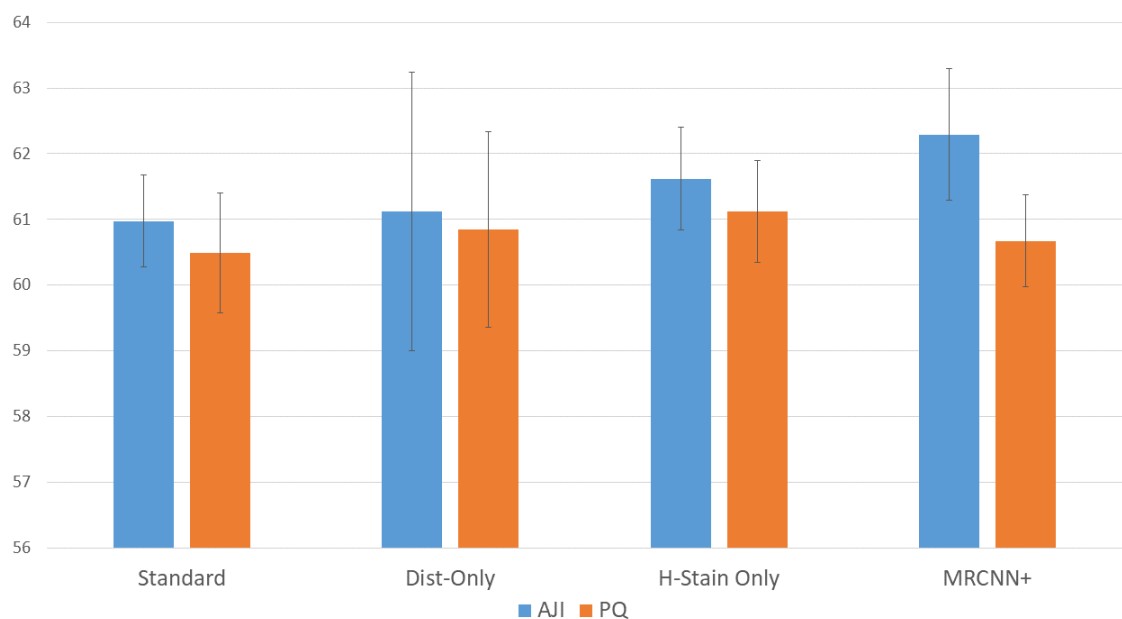

Figure 5: Ablations of additional information with standard deviation across runs.

Figure 5 shows ablations to derive the use of additional information in the form of extracted H-stain and estimated distance information.

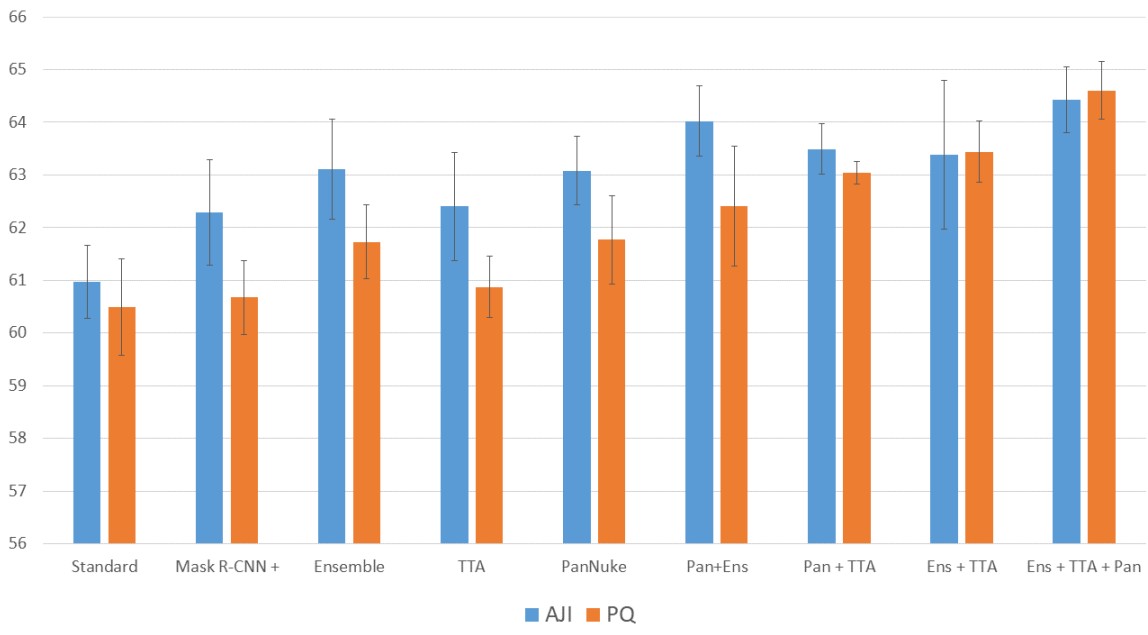

Figure 6: Graphical results of ablations of inference improvements across runs.

Figure 6 shows the results of the 5 runs in a chart with standard deviation across runs. Figure 7 shows sample results of inferences on the validation set.

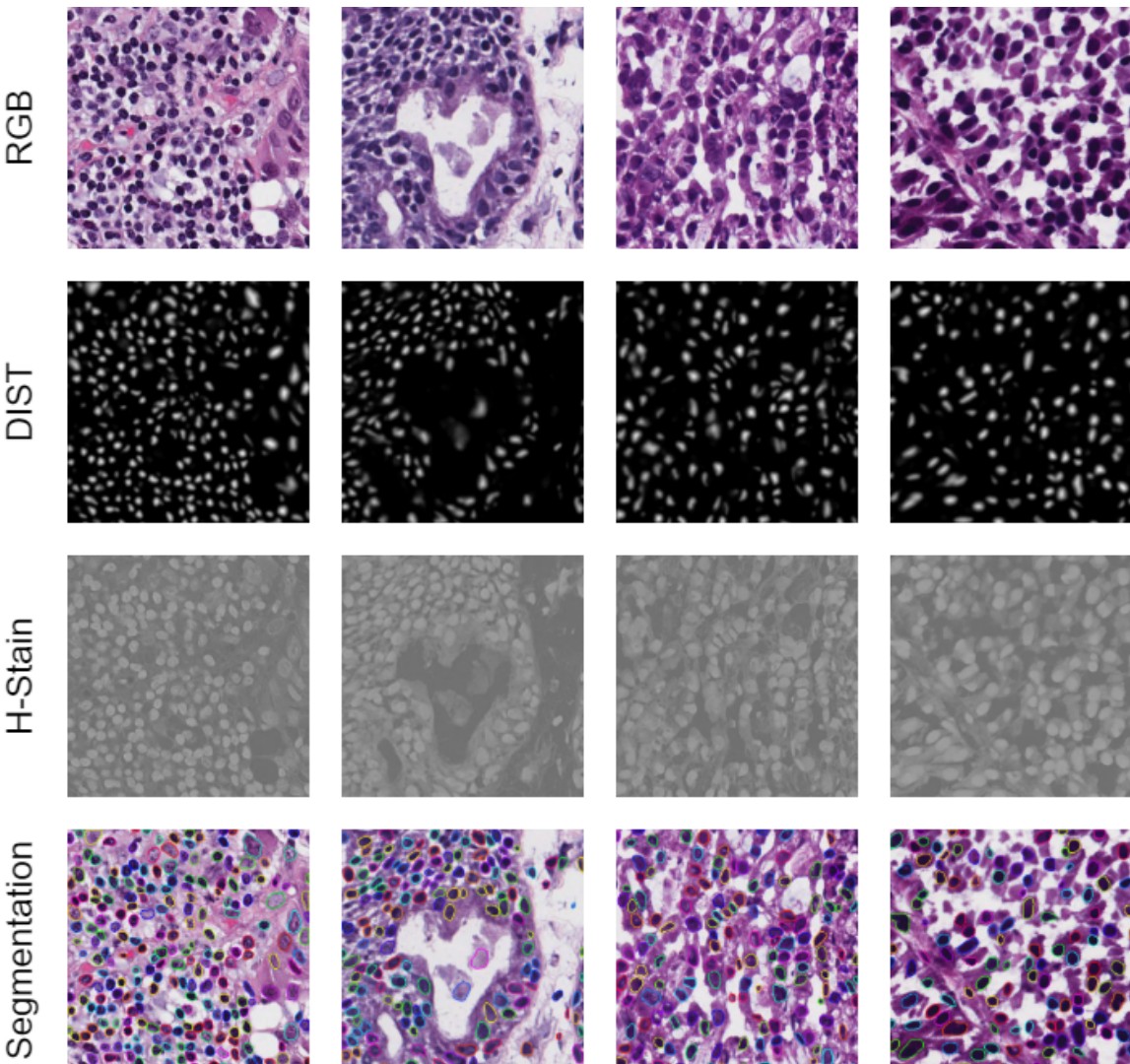

Figure 7: Pre-training and fine-tuning scheme.

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
