# OpenReview forum: "Improving Mask R-CNN for Nuclei Instance Segmentation in Hematoxylin & Eosin-Stained Histological Images"
_MICCAI.org/2021/Workshop/COMPAY — COMPAY 2021_

### Official Review · Reviewer_oHND · 2021-08-09
**Improved method for nuclei segmentation using a modified Mask-RCNN**

**Rating:** 6
**Confidence:** 4

**Review:**

In their study, the authors present an improved version of Mask-RCNN for nuclear segmentation and overlap separation.
Specifically, they show that segmentation quality can be improved by extending Mask-RCNN to also process H-stain entensity and distance maps, as well as using pre-training and test-time augmentation.
The paper is clearly written, and the code made publicly available to the community.

I suggest the following points to be addressed in a revised version:
-The authors use H-stain intensity and euclidean distance maps as additional inputs into the Mask-RCNN. What motivated this particular choice? Did the authors also try other image transformations?
-The overall improvement reported due to the modifications made in the paper is about 3.5 percentage points in segmentation score. How significant is this improvement? Can it be reproduced on external validation data?
How does this improvement compare to the annotation inter- and intra-observer variability the authors mention?

As a minor comment, segmentations in Fig.3 are hardly visible. The outlines should be drawn more clearly.

---

### Official Review · Reviewer_CVwa · 2021-08-19
**Modified Mask R-CNN improves nuclei segmentation in H&E**

**Rating:** 6
**Confidence:** 4

**Review:**

The paper presents a method based on Mask R-CNN, where additional features have been added to improve nuclei segmentation performance, namely two additional input channels (H and E, after color deconvolution), test time data augmentation, model ensembling, and different pre-training strategies.
The authors show that adding any of these features improves the performance of the model when looking at one (or more) of the performance metrics considered in this study.
The authors aim at making the code publicly available.

The main results of this study are summarized in Table 1, where an ablation study is performed to show the contribution of each additional feature or a combination of features. The authors highlight the combination that achieves AJI = 64.45 as the best overall solution, but does not explain why AJI is the best overall metric to consider to say so; for example, the last configuration in the table achieves a PQ = 64.06, which is higher (although probably not statistically different) than the PQ of the highlighted solution. I wonder how a combination can be defined as the "best", also because most results are very close and probably not significantly different. I think it would be useful to complement Table 1 with visual results of cell segmentation using different methods and highlight the differences.

The Distance U-Net model is mentioned and used, but I think its explanation is not very clear in the paper, it is not clear where and how it is used, it would be good to make a figure showing a schematic representation of the entire pipeline, for clarity, and show the role of distance U-Net.

It would also be interesting to compare (RGB + H + E) with (H + E) only, as most of the information is in the H and E channel.

---

### Decision · Program_Chairs · 2021-08-25

Accept